# *In vitro* and *in silico* studies for the identification of anti-cancer and antibacterial peptides from camel milk protein hydrolysates

**Mohammad Javad Taghipour[1], Hamid Ezzatpanah ORCID[1]\*, Mohammad Ghahderijani[2]**

**1** Department of Food Science and Technology, Science and Research Branch, Islamic Azad University, Tehran, Iran, **2** Department of Agricultural Systems Engineering, Science and Research Branch, Islamic Azad University, Tehran, Iran

\* hamidezzatpanah@srbiau.ac.ir

**Data Availability Statement:** All relevant data are within the manuscript and its Supporting information files.

## Abstract

Today, breast cancer and infectious diseases are very worrying that led to a widespread effort by researchers to discover natural remedies with no side effects to fight them. In the present study, we isolated camel milk protein fractions, casein and whey proteins, and hydrolyzed them using pepsin, trypsin, and both enzymes. Screening of peptides with anti-breast cancer and antibacterial activity against pathogens was performed. Peptides derived from whey protein fraction with the use of both enzymes showed very good activity against MCF-7 breast cancer with cell viability of 7.13%. The separate use of trypsin and pepsin to digest whey protein fraction yielded peptides with high antibacterial activity against *S. aureus* (inhibition zone of 4.17 ± 0.30 and 4.23 ± 0.32 cm, respectively) and *E. coli* (inhibition zone of 4.03 ± 0.15 and 4.03 ± 0.05 cm, respectively). Notably, in order to identify the effective peptides in camel milk, its protein sequences were retrieved and enzymatically digested *in silico*. Peptides that showed both anticancer and antibacterial properties and the highest stability in intestinal conditions were selected for the next step. Molecular interaction analysis was performed on specific receptors associated with breast cancer and/or antibacterial activity using molecular docking. The results showed that P3 (WNHIKRYF) and P5 (WSVGH) peptides had low binding energy and inhibition constant so that they specifically occupied active sites of protein targets. Our results introduced two peptide-drug candidates and new natural food additive that can be delivered to further animal and clinical trials.

## Introduction

Research has shown that breast cancer is associated with infectious diseases [1]. According to the WHO reports [2], breast cancer is the most prevalent cancer in the world with about 7.8 million women diagnosed by the end of 2020. This organization has also informed that 55% of global deaths have occurred by infectious diseases in 2019. These alarming statistics have prompted researchers around the world to look for a way to control them. On condition that the breast cancer is identified in the early stages, the treatment can achieve a survival possibility

**Funding:** The author(s) received no specific funding for this work.

**Competing interests:** The authors have declared that no competing interests exist.

of 90% or higher. For this, the use of anticancer drugs including chemotherapy, endocrine therapy, and targeted biologic therapy such as peptides, antibodies, and secondary metabolites are very common.

The worldwide sales of peptide drugs were reached to US$ 70 billion in 2019 [3], which reveals a lot of attention has been paid to peptide-based drugs. Milk is a source of bioactive peptides which are inactively located within the parent protein sequences [4, 5]. These peptides are released from parental protein sequences using hydrolysis and cause a variety of biological activities, including antiviral [6], opioid [7], antithrombotic [8], antihypertensive [9], and anti-oxidant [10, 11]. They provide other benefits in comparison with proteins, including being small in size and more easily passing through the digestive tract, low osmotic pressure, low antigenicity, good taste, easy synthesis, and easily recombinant production [12].

Previous researches reported antibacterial and anticancer peptides from mammalian milk proteins by hydrolysis using various enzymes, including pepsin, trypsin, chymosin, chymo-trypsin [13, 14]. The hydrolysates of camel milk proteins have exhibited some other biological activities, including antioxidant, antihypertensive, antidiabetic, anti-inflammatory, and antic-holesterolemic [15–18]. It seems that, camel milk is an excellent and interesting food source can provide therapeutic peptides which will contribute to both nutrition and human health through use as natural ingredients in food or drugs [16, 19]. Consumption of camel milk is common among peoples because of its similarity to human milk [20]. Camel milk is consisted two main protein fractions including casein; αS1-, αS2-, β-, and κ-casein, and whey proteins; α-lactalbumin, serum albumin, lactoferrin, lactophorin, immunoglobulins, some peptides and lacks of β-lactoglobulin [20].

Peptide screening and computational biology will support the discovery of drugs. However, new methods must be complemented by laboratory studies in order to provide results that are more accurate. It would be very promising to find anticancer drugs that cannot only control the progression of cancer but also subsequent infectious diseases. The aim of this study was to investigate and identify various anti-breast cancer and antibacterial peptides from different camel milk protein fractions. Firstly, we studied their biological properties *in vitro* and then identified effective peptides using bioinformatics tools, followed by analyzing physicochemical properties and their binding to specific receptors using molecular docking.

## Materials and methods

### *In vitro* protein preparation

Camel milk (*Camelus dromedarius*) was purchased from the local market in Shiraz, Iran, followed by stored at -20 ˚C. The casein and serum proteins were fractioned according to the previously described method [21] with some modifications. Briefly, camel milk was defatted using a centrifuge at 4255 ×g at 4 ˚C for 15 min. Afterward, the pH of defatted milk was adjusted to 4.6 (isoelectric pH of casein) using 1N HCl and then allowed to settle at 4 ˚C for 30 min followed by centrifugation at 10000 ×g at 4 ˚C for 15 min. The precipitated casein fraction was collected and washed twice with $ddH_2O$. The supernatant was again centrifuged at the above condition for 60 min and supernatant, as whey protein fraction, was collected. The pH of casein and whey protein fractions was adjusted to 7.0 and afterward lyophilized and stored at -20 ˚C for the next analysis.

### *In vitro* enzymatic digestion

Lyophilized proteins were dissolved in double distilled water and enzymatic digestion was performed using pepsin, trypsin, and both in the optimum conditions. These enzymes were selected based on gastrointestinal simulated condition. Briefly, hydrolysis was conducted in

the ratio of 1:25 enzyme to protein at 37 ˚C and pH of 2.5 for 12 h using pepsin, whereas it was in the same conditions with a pH of 7.5 using trypsin. In the combination of both enzymes, the first hydrolysis was conducted using pepsin. After the inactivation of pepsin, the second hydrolysis was conducted using trypsin. Finally, each reaction was heated at 95 ˚C for 20 min to inactivate enzyme and the samples were centrifuged at 12000 ×g for 30 min followed by collecting the supernatant. Hydrolysates were lyophilized and stored at -20 ˚C for further analysis.

## Determining the hydrolysis degree

Hydrolysis degree of hydrolysates over time was determined according to the previously described method [22]. Briefly, sampling was conducted at 0, 1, 2, 4, 6, and 12 h after hydrolysis and the content of α-amino nitrogen was determined using the OPA method. The degree of hydrolysis was calculated using the following equations:

$$\text{Serine-}NH_2 = \left(OD_{sample} - OD_{blank}\right)/(OD_{standard} - OD_{blank}) \times 0.9516 \times 0.1 \times 100/X \times P$$

$$h = (\text{serine-}NH_2 - \beta)/\alpha \text{ meqv/g protein}$$

$$DH(\%) = h/h_{tot} \times 100$$

Whereas X and P are gram of sample and the percentage of protein in sample, respectively. In addition, α, β, and $h_{tot}$ are equal to 1.039, 0.383, and 8.2 for casein and 1.00, 0.40, and 8.8 for whey proteins.

## Antibacterial activity

The antibacterial activity of hydrolysates was evaluated using the agar well diffusion method according to the described method elsewhere [23] with minor alterations. Briefly, four bacterial strains of *Klebsiella pneumoniae* (PTCC 1053), *Pseudomonas aeruginosa* (PTCC 1074), *Staphylococcus aureus* (PTCC 1112), and *Escherichia coli* (PTCC 1330) were cultured in the liquid LB medium overnight at 37 ˚C. Afterward, each strain was prepared at 0.5 McFarland standard. For antibacterial screening, 1 mL of bacteria inoculum was mixed with solid LB medium and poured into the petri dish. After solidification, the wells were created and 150 μL of each hydrolysate at 400 μg/mL was pipetted into each well. The plates were incubated at 4 ˚C for diffusing followed by at 37 ˚C for 18 h. The inhibition potency of each hydrolysate was measured using the diameter of appeared halo.

## Anticancer activity

The anticancer activity of hydrolysates was evaluated using the MTT ((3-[4,5-dimethylthiazol-2-yl]-2,5 diphenyl tetrazolium bromide)) assay according to the previously reported methods [24, 25]. For this purpose, the MCF-7 breast cancer cell line was maintained in RPMI-1640 medium at 37 ˚C and 5% $CO_2$ in a humid atmosphere. Thereafter, $1 \times 10^4$ cells were seeded into each well of the 96-well plate and incubated at 37 ˚C and 5% $CO_2$ for 24 h. Afterward the medium was removed and hydrolysates diluted in the medium at 400 μg/mL followed by add on plate being incubated under the above conditions for 24 h. The treatments were removed and MTT dye prepared in medium (0.5 mg/mL/well) was added. The plate was again incubated under the above conditions for 4 h and then MTT dye was removed. The cells were treated with DMSO (dimethyl sulfoxide) and incubated under the above conditions for 15 min. Finally, the absorbance was measured at 570 nm using a microplate reader. The cell

viability percentage was calculated by the described equation elsewhere [26] and $IC_{50}$ value was calculated.

## Statistical analysis

All experiments were done in triplicates and results were expressed as mean ± SD. Statistical analysis and mean comparison were performed in the form of a completely random design using SAS 9.4 (Institute Inc., Cary, NC, USA) and the Tukey test, respectively. Drawing graphs and calculating the $IC_{50}$ value were done using GraphPad Prism software.

## Protein sequences

The protein sequences of the milk of camel (*Camelus dromedarius*), including alpha-S1-casein (accession number: O97943), alpha-S2-casein (O97944), beta-casein (Q9TVD0), kappa-casein (L0P3Z7), albumin (A0A5N4EFP0), and alpha-lactalbumin (A0A5N4DFK4), were retrieved from the UniProt database (https://www.uniprot.org/).

## *In silico* enzymatic digestion and screening

The milk protein sequences were applied to digest using the Peptide Cutter Server (http://www.expasy.ch/tools/peptidecutter/) with pepsin and trypsin enzymes individually (S1 Table in S1 File). Derived peptides were screened for anticancer and antimicrobial activities using the AntiCP 2.0 server (https://webs.iiitd.edu.in/raghava/anticp2) with SVM threshold of 0.6 and both methods of Model 1 and 2, and the Antimicrobial Peptide Scanner vr.2 (https://www.dveltri.com/ascan/v2/ascan.html) server with probabilities of 0.6, respectively. Afterward, the half-life of peptides that showed both anticancer and antimicrobial activity was evaluated in intestinal conditions using the HLP server (http://crdd.osdd.net/raghava/hlp/pep_both.htm) with SVM based model, followed by peptides with high half-life and high stability selected for further analysis (Table 1).

## Analysis of physicochemical properties

To raise our knowledge of a better understanding of the derived peptides (Table 1), their physicochemical properties, including the molecular weight, theoretical pI, amino acid composition, the number of negatively charged residues (Asp + Glu), the number of positively charged

**Table 1.  The main peptides used for further analysis in this study.**

| Number | Source | Peptide sequence | AntiCP score | | AntiMP score | Half-life (sec.) | Stability |
|---|---|---|---|---|---|---|---|
| | | | Model 1 | Model 2 | | | |
| P1 | alpha-S1-casein | ARPKY | < 0.60 | 0.61 | 0.63 | 1.36 | High |
| P2 | alpha-S1-casein | LIPRVKL | 0.64 | 0.66 | 0.75 | 1.87 | High |
| P3 | alpha-S2-casein | WNHIKRYF | 0.72 | < 0.60 | 0.87 | 1.43 | High |
| P4 | alpha-S2-casein | FFIFTCLLAVVLAK | 0.62 | 0.69 | 0.99 | 1.10 | High |
| P5 | albumin | WSVGH | 0.77 | < 0.60 | 0.66 | 1.27 | High |
| P6 | albumin | LGRVGTKCCTL | < 0.60 | 0.68 | 0.88 | 1.32 | High |
| P7 | albumin | AACLLPK | 0.81 | 0.66 | 0.82 | 2.68 | High |
| P8 | alpha-lactalbumin | FTKCKL | < 0.60 | 0.73 | 0.88 | 2.50 | High |

*Note*: AntiCP and AntiMP scores indicated the scores retrieved from the AntiCP and Antimicrobial Peptide Scanner servers, respectively. Half-life indicates the half-life of peptides in the intestinal conditions.

residues (Arg + Lys), atomic composition, total number of atoms, aliphatic index, and grand average of hydropathicity (GRAVY), were analyzed using the ProtParam tool (https://web.expasy.org/protparam/).

## Receptor and ligand preparation

Three-dimensional structure model of peptides presented in Table 1 was predicted using the PEP-FOLD 3.5 server (https://mobyle.rpbs.univ-paris-diderot.fr/cgi-bin/portal.py#forms::PEP-FOLD3). According to the mode of action of antimicrobial agents in the inactivation of bacteria, we selected four receptors including penicillin-binding protein 1a (PBP1a: as inhibitors of cell wall synthesis), isoleucyl-tRNA synthetase (IARS: as inhibitors of protein synthesis), DNA gyrase (as inhibitors of nucleic acid synthesis), dihydrofolate reductase (DHFR: as antimetabolites) and their crystal structure obtained from the Protein Data Bank (https://www.rcsb.org/) with PDB entry of 3UDI, 1JZQ, 3TTZ, and 3SRW, respectively. In addition, the crystal structure of two receptors, which overexpress in breast cancer including Myeloid Cell Leukemia 1 (MCL-1) and estrogen alpha (ERα), were obtained from the Protein Data Bank with PDB entry of 5FDO and 1G50, respectively. All receptors were introduced into the Auto-Dock Tools (the Scripps Research Institute, La Jolla, CA, USA) software and prepared for docking by adding AD4 type atoms, removing all water molecules except A-2 from 3TTZ and 128 from 3SRW, which are essential for correct docking pose, and adding Kollman charge and polar hydrogens. Afterward, they were used for molecular docking.

## Molecular docking and visualization

Molecular docking was performed using the AutoDock Vina 1.1.2 software (the Scripps Research Institute, La Jolla, CA, USA). The active site residues of each receptor were identified from previously reported literature [27–33] and the grid box was set on these residues followed by each analysis was independently run three times. The results were presented as the binding energy (Kcal/mol) and the inhibition constant (Ki) [33]. The inhibition constant was calculated from the binding energy according to the following equation: $\Delta G = RT ln Ki$, whereas $\Delta G$, R, and T are binding energy (cal/mol), gas constant (1.987 cal/K mol), and temperature (310.15 K), respectively. After docking analysis, the receptor-ligand complexes were visualized using the PyMol software to illustrate interactions.

## Results and discussion

### Hydrolysis degree

The measurement of hydrolysis degree in proteolytic reactions is one of the important methods to assess the correctness of the reaction. Hence, we determined this factor over time using a method described in the material and method section [22]. The results showed that the hydrolysis degree increased by increasing the time (Fig 1). At time 0, no hydrolysis has taken place, so that the degree of hydrolysis of the parent proteins varied from 1.9 to 2.8% and increased over time with the addition of enzymes (Fig 1). It indicates that enzymatic digestion was done and derived peptides were evaluated for their probable antibacterial and anticancer activities. By the way, it should be noted that the hydrolysis degree of casein and whey proteins fractions was more than others when both enzymes were used (Fig 1).

### Antibacterial activity

Bioactive peptides are used as alternative drugs or food preservative due to their important role in innate immunity and antimicrobial activity. Today, they are utilized as significant

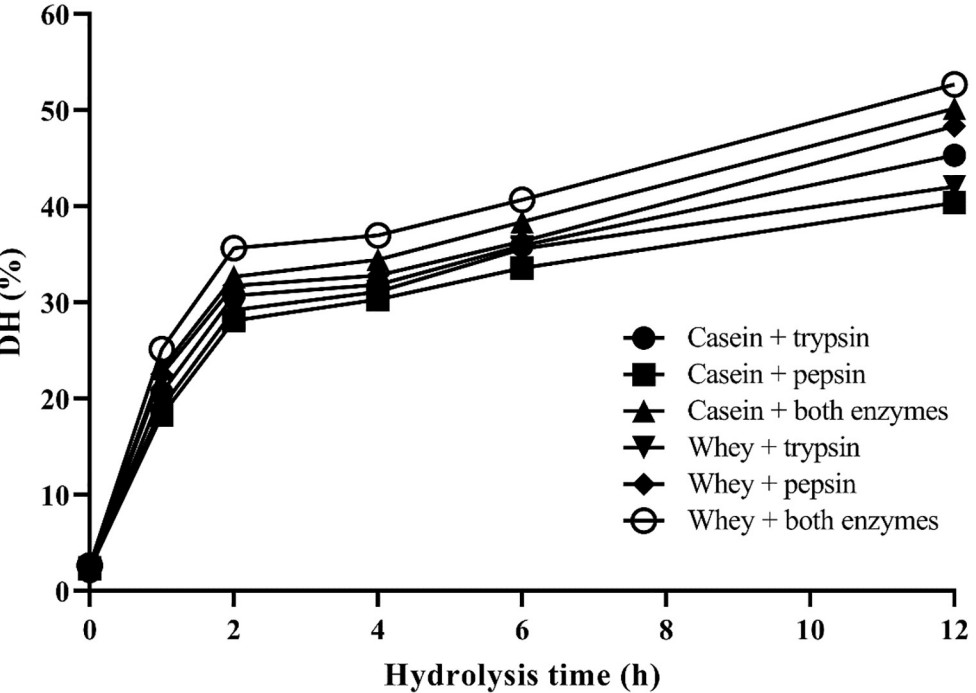

**Fig 1. Determining the hydrolysis degree of hydrolysates over time.** Data are expressed as the mean of a triplicate experiment.

ingredients in functional foods, pharmaceuticals, and nutraceuticals. We evaluated the anti-bacterial activity of camel milk protein-derived peptides to introduce potential source of food additive or drug candidates. Regarding casein hydrolysis, the highest antibacterial activity was observed in casein-derived peptides by trypsin, showed the highest antibacterial activity against *K. pneumoniae* and *S. aureus* (Table 2). By the way, the hydrolysis of casein by both enzymes was reduced the antibacterial potency of hydrolysates in comparison with casein hydrolysis by trypsin, which could be due to more digestion of actual antibacterial peptides. On the other hand, casein hydrolysis by pepsin was produced the best antibacterial peptides against *E. coli*, whereas casein hydrolysis by both enzymes revealed a good antibacterial effect

**Table 2. Evaluation of antibacterial activity of protein hydrolysates (400 µg/ml) derived by various enzymes.**

| Protein | Enzyme | *K. pneumoniae* | *P. aeruginosa* | *S. aureus* | *E. coli* |
|---------|--------|-----------------|-----------------|-------------|-----------|
| Casein | None | 0.0[c] | 0.0[b] | 0.0[d] | 0.0[c] |
| | Trypsin | 2.77 ± 0.25[a] | 1.32 ± 0.63[ab] | 3.10 ± 0.10[a] | 1.07 ± 0.30[b] |
| | Pepsin | 0.77 ± 0.25[b] | 1.4 ± 0.53[ab] | 2.30 ± 0.30[b] | 2.37 ± 0.32[a] |
| | Pepsin + Trypsin | 2.13 ± 0.15[a] | 2.03 ± 0.15[a] | 1.03 ± 0.15[c] | 0.73 ± 0.25[bc] |
| Whey | None | 0.0[b] | 0.0[b] | 0.0[c] | 0.0[c] |
| | Trypsin | 0.93 ± 0.30[a] | 1.0 ± 0.10[a] | 4.17 ± 0.30[a] | 4.03 ± 0.15[a] |
| | Pepsin | 0.90 ± 0.26[a] | 1.03 ± 0.25[a] | 4.23 ± 0.32[a] | 4.03 ± 0.05[a] |
| | Pepsin + Trypsin | 0.87 ± 0.15[a] | 0.90 ± 0.26[a] | 2.03 ± 0.25[b] | 1.93 ± 0.20[b] |

*Note*: The numbers are the size of inhibition zone (cm). Letters separately indicates a significant level at *P-value* ≤ 0.01 in each group. Data are expressed as mean ± SD from two independent triplicate experiments.

against *P. aeruginosa* in comparison with separately used enzymes. Intact casein fraction showed no antibacterial activity.

Non-hydrolyzed whey protein fraction showed no antibacterial activity, but hydrolysis by trypsin or pepsin revealed antibacterial activity against the *S. aureus* and *E. coli* better than casein-derived peptides (Table 2). Although these peptides showed antibacterial activity against *K. pneumoniae* and *P. aeruginosa*, no significant difference was observed between trypsin, pepsin, and both enzymes hydrolysis. A previous study has showed that hydrolysis of casein and whey proteins from Etawa goat milk by bromelain enzyme is able to inhibit the growth of *S. aureus* and *E. coli* with a maximum inhibition zone of 2.68 and 3.05 mm, respectively [34]. Furthermore, Jrad et al. [35] showed that hydrolysis of dromedary lactoferrin using pepsin led to an increase in the growth inhibition of *Listeria innocua* by 28.4% at 0.5 mg/mL compared to unhydrolyzed lactoferrin (inhibition percentage of < 10% at 1 mg/mL). Apart from the direct use of proteases, microbial fermentation of camel milk proteins leads to the production of antibacterial peptides. Hence, Muhialdin and Algboory [36] identified 23 peptides from the hydrolysates of different camel milk proteins fermented by *Lactobacillus plantarum*, which were able to exhibit antibacterial activity against *S. faecalis*, *Shigella dysenteriae*, *S. aureus*, and *E. coli*. Another study has indicated that camel whey proteins hydrolysis by trypsin at 40 ˚C for 4 h has ability to killing *S. aureus*, *E. coli*, *Salmonella typhimurium*, and *Streptococcus mutans* with an inhibition zone of 11.90, 14.53, 8.36, and 5.60 mm, respectively [37]. Recent study revealed that the temperature and time used to hydrolysis can affect the generation of antibacterial peptides in comparison with our work at 37 ˚C for 12 h. Therefore, optimization of hydrolysis conditions by each enzyme is essential to achieve the maximum effect. We, in turn, can say that hydrolysis of camel milk proteins, depending on the type of target bacterium; can be good candidates for developing new peptide-based drugs or natural food preservative.

## Anticancer activity

For the time being, finding the emerging solutions for tackling breast cancer and consequential infectious diseases is very important. Therefore, the use of hydrolysates of food proteins as anticancer agents is a good option for the development of peptide-based drugs due to their lack of side effects since several anticancer peptide-based drugs have been approved [38]. Accordingly, we evaluated the anticancer activity of camel milk protein hydrolysates against the MCF-7 cell line. The results showed that hydrolysis of whey proteins reveals an anticancer activity better than casein counterparts (Fig 2). When whey protein hydrolysates derived from both enzymes were tested at 400 μg/mL, the cell viability percentage was decreased to 7.13% in comparison with intact whey proteins (74.23%). It was assumed that the isolation of effective peptides has the potential to become a potent peptide-based drug. Otherwise, the casein protein hydrolysates showed anticancer activity, i.e., a cell viability percentage of 57.46%, 63.11%, and 72.42% for trypsin-, pepsin-, and both enzymes (pepsin and trypsin)-derived hydrolysates, respectively, where their inhibition potency was not comparable with whey proteins hydrolysates. On the other hand, whey and casein proteins (parent proteins that used as control) respectively killed approximately 30% and 10% of total cancerous cells, which is lower than anticancer activity of hydrolysates (Fig 2). Although, Homayouni-Tabrizi and co-workers showed that the isolated peptides from hydrolysates derived from camel whey proteins by pepsin and pancreatin revealed no significant anticancer activity against the HepG2 cell line [39]. Previous studies have shown that camel whey protein hydrolysates by pepsin (P-5.2) reveal a good anticancer activity against the HCT116 cell line with an $IC_{50}$ of 221 μg/mL [40]. Also, it is reported that the isolated peptides from Buffalo whey proteins are able to inhibit the growth of the Caco-2 cell line at 800 μg/mL for 24 h [41]. Therefore, our results in combination with

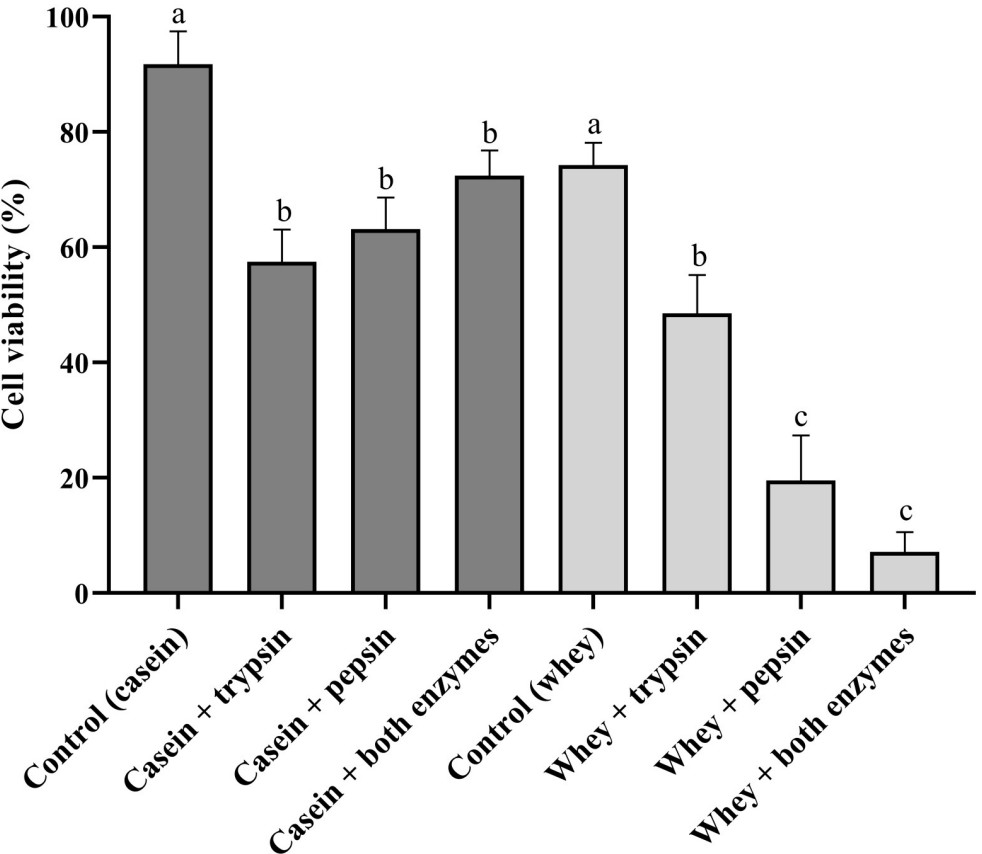

**Fig 2. Anticancer activity of hydrolysates against MCF-7 breast cancer cell line using the MTT assay method.** Data are expressed as mean ± SD from two independent triplicate experiments. Letters indicate a significant level at *P-value* ≤ 0.01 for comparing each hydrolysate with its control.

previous studies have provided a good foundation for further study into camel milk protein hydrolysates and their ability to become a potential peptide-based therapeutic agent for cancer.

### *In silico* screening

The screening of bioactive compounds using computer is a powerful, cost-effective, and short time method. Consequently, we retrieved the protein sequences in camel's milk from the Uni-Prot database and then digested them in silico using the pepsin and trypsin enzymes individually. After digestion, about 374 peptides were derived and used for screening based on the potency of their antimicrobial and anticancer activities (S1 Table in S1 File). Among them, nine peptides showed both anticancer and antimicrobial activity with a probability more than 0.6. These peptides evaluated based on their stability in intestinal condition and eight of them showed high stability (Table 1). Afterward in order to better understand the properties affecting the potency of peptides, their physicochemical properties were investigated.

### Physicochemical properties

Physicochemical properties of peptides can reveal the potential of their bioactivity. Hence, evaluating these features is a shortcut to the final aims. The linear antimicrobial peptides,

including anticancer peptides, are positively charged and abundant in histidine and hydrophobic amino acids, including phenylalanine, isoleucine, leucine, and tryptophan, which is an inherent feature of them for facilitating α-helix formation in the membrane [42, 43]. Thus, P3 and P4 were rich in these residues. Furthermore, P5 was rich in histidine and tryptophan residues. This indicates that these peptides are more effective against bacteria or cancer in comparison with other peptides, including P1, P2, P6, P7, and P8 (Table 3).

On the other hand, earlier literature has indicated that the pore formation by peptides happens in hydrophobicity thresholds ranging from—0.3 to 0 kcal/mol, and a net charge threshold of 2 [44]. According to this, P5 with a GRAVY score of—0.22 may reveal a bactericidal effect

**Table 3. The physicochemical properties of desired peptides using the ProtParam tool.**

| Physicochemical properties | P1 | P2 | P3 | P4 | P5 | P6 | P7 | P8 |
|---|---|---|---|---|---|---|---|---|
| **Amino acid composition (% w/w)** | | | | | | | | |
| Ala (A) | 20.0 | 00.0 | 00.0 | 14.3 | 00.0 | 00.0 | 28.6 | 00.0 |
| Arg (R) | 20.0 | 14.3 | 12.5 | 00.0 | 00.0 | 09.1 | 00.0 | 00.0 |
| Asn (N) | 00.0 | 00.0 | 12.5 | 00.0 | 00.0 | 00.0 | 00.0 | 00.0 |
| Asp (D) | 00.0 | 00.0 | 00.0 | 00.0 | 00.0 | 00.0 | 00.0 | 00.0 |
| Cys (C) | 00.0 | 00.0 | 00.0 | 07.1 | 00.0 | 18.2 | 14.3 | 16.7 |
| Gln (Q) | 00.0 | 00.0 | 00.0 | 00.0 | 00.0 | 00.0 | 00.0 | 00.0 |
| Glu (E) | 00.0 | 00.0 | 00.0 | 00.0 | 00.0 | 00.0 | 00.0 | 00.0 |
| Gly (G) | 00.0 | 00.0 | 00.0 | 00.0 | 20.0 | 18.2 | 00.0 | 00.0 |
| His (H) | 00.0 | 00.0 | 12.5 | 00.0 | 20.0 | 00.0 | 00.0 | 00.0 |
| Ile (I) | 00.0 | 14.3 | 12.5 | 07.1 | 00.0 | 00.0 | 00.0 | 00.0 |
| Leu (L) | 00.0 | 28.6 | 00.0 | 21.4 | 00.0 | 18.2 | 28.6 | 16.7 |
| Lys (K) | 20.0 | 14.3 | 12.5 | 07.1 | 00.0 | 09.1 | 14.3 | 33.3 |
| Met (M) | 00.0 | 00.0 | 00.0 | 00.0 | 00.0 | 00.0 | 00.0 | 00.0 |
| Phe (F) | 00.0 | 00.0 | 12.5 | 21.4 | 00.0 | 00.0 | 00.0 | 16.7 |
| Pro (P) | 20.0 | 14.3 | 00.0 | 00.0 | 00.0 | 00.0 | 14.3 | 00.0 |
| Ser (S) | 00.0 | 00.0 | 00.0 | 00.0 | 20.0 | 00.0 | 00.0 | 00.0 |
| Thr (T) | 00.0 | 00.0 | 00.0 | 07.1 | 00.0 | 18.2 | 00.0 | 16.7 |
| Trp (W) | 00.0 | 00.0 | 12.5 | 00.0 | 20.0 | 00.0 | 00.0 | 00.0 |
| Tyr (Y) | 20.0 | 00.0 | 12.5 | 00.0 | 00.0 | 00.0 | 00.0 | 00.0 |
| Val (V) | 00.0 | 14.3 | 00.0 | 14.3 | 20.0 | 09.1 | 00.0 | 00.0 |
| **Molecular weight (Da)** | 633.75 | 838.1 | 1163.35 | 1585.02 | 584.63 | 1150.42 | 714.92 | 738.94 |
| **Theoretical pI** | 9.99 | 11 | 9.99 | 8.22 | 6.74 | 8.96 | 8.27 | 9.31 |
| **Number of negatively charge residues** | 0 | 0 | 0 | 0 | 0 | 0 | 0 | 0 |
| **Number of positively charge residues** | 2 | 2 | 2 | 1 | 0 | 2 | 1 | 2 |
| **Atomic composition** | | | | | | | | |
| C | 29 | 40 | 57 | 80 | 27 | 47 | 32 | 34 |
| H | 47 | 75 | 78 | 125 | 36 | 87 | 58 | 58 |
| N | 9 | 11 | 16 | 15 | 8 | 15 | 8 | 8 |
| O | 7 | 8 | 11 | 16 | 7 | 14 | 8 | 8 |
| S | 0 | 0 | 0 | 1 | 0 | 2 | 1 | 1 |
| **Total number of atoms** | 92 | 134 | 162 | 237 | 78 | 165 | 107 | 109 |
| **Aliphatic index** | 20 | 208.57 | 48.75 | 167.14 | 58 | 97.27 | 140 | 65 |
| **GRAVY** | - 1.9 | 0.9 | - 1.25 | 2.44 | - 0.22 | 0.56 | 1.17 | 0.1 |

*Note*: The negatively and positively charge residues are included Asp + Glu and Arg + Lys, respectively.

GRAVY: Grand average of hydropathicity.

by using pore formation or a mode of action related to membrane disorders. Among all, P1 and P3 had a net charge of + 2 and were the most hydrophobic (Table 3). Also, it is indicated that all peptides rich in particular amino acids, such as arginine, tryptophan, proline, glycine, cysteine, and histidine reveal antimicrobial activity, as seen in this study (Table 3) [42].

In the case of peptides that contain histidine, such as P3 and P5, histidine is an opportune residue to control pH and cationic ion sensing for facilitating cell penetration, so that its targets also may be intracellular. In addition, the presence of histidine causes the formation of stable non-covalent complexes with acidic residues such as phosphate groups [42]. Generally, the modes of action and potency of antimicrobial peptides determine by some of these physico-chemical properties, which is highlighted in each peptide.

## Molecular docking analysis

Molecular docking, one of the most fundamental and noteworthy strategies in drug discovery, studies the binding mode of a ligand in the active site of proteins [45]. To evaluate the mode of action of desired peptides, we performed molecular docking analysis using AutoDock Vina and showed the bound state, binding energy, and inhibition constant of them against various proteins involved in growth and development of bacteria or breast cancer. Accordingly, all peptides showed a good binding mode to protein targets (S1-S6 Figs in S1 File), so that some of them made a more specific interaction. For instance, P4 just made an interaction with DNA gyrase protein with a binding energy less than others did.

In a closer look, P3 formed hydrogen bonds with His1377, Phe1461, Leu1462, Ser1464, Ser1468, Arg1548, Arg2412, and Asp2426 residues of ERα protein with a binding energy of–7.1 kcal/mol and inhibition constant of 9.9 μM (Fig 3a, S2 Table in S1 File). Although none of the P3-interacting residues contained residues in the active site of ERα, it is suspected that the P3 may have acted as an allosteric ligand. In contrast, this peptide exactly interacted with the

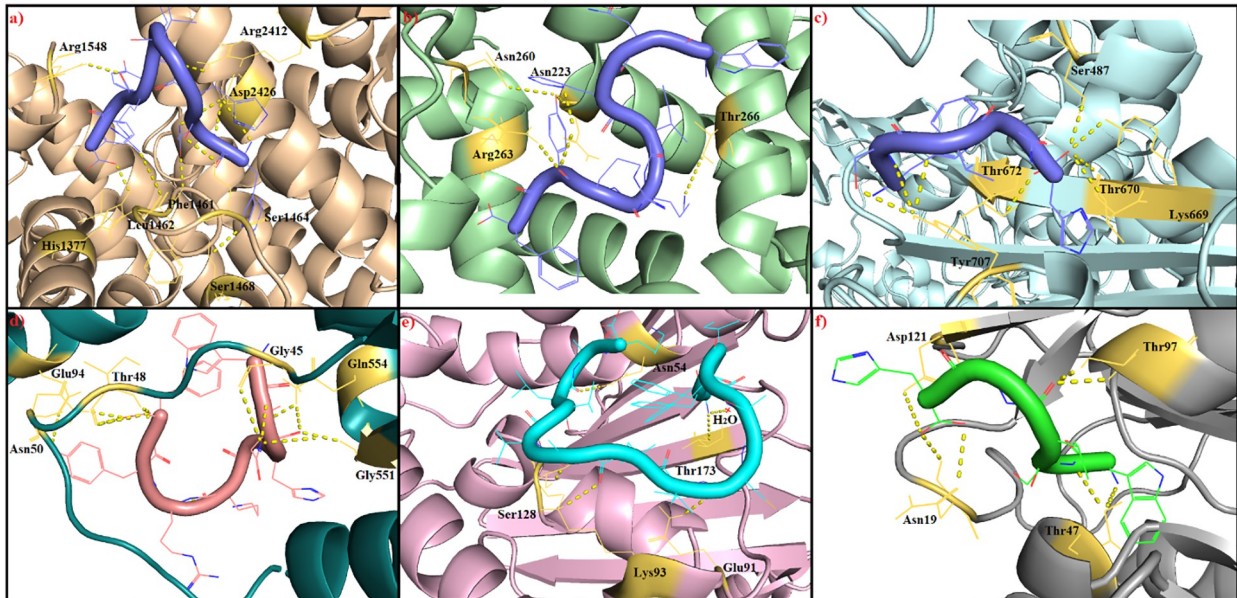

**Fig 3. Molecular interactions of the best peptide-receptor complexes based on the binding energy from the molecular docking analysis.** a) P3-ERα complex, b) P3-MCL-1 complex, c) P5-PBP1a complex, d) P3-IARS complex, e) P4-DNA gyrase complex, and f) P5-DHFR complex. All hydrogen bonds have shown in dashed lines with yellow color.

residues in the binding site of MCL-1 protein, including Asn260, Arg263, and Thr266, with a binding energy of– 7.2 kcal/mol and inhibition constant of 8.5 μM (Fig 3b, S2 Table in S1 File). It shows that P3 acts as an agonist or antagonist ligand against MCL-1. Therefore, P3 can appear different mechanisms to suppress breast cancer.

In the case of the antibacterial activity of desired peptides, P3, P4, and P5 showed the ability to interact with various targets involved in the growth and development of bacteria. PBP1a is a membrane-associate protein that synthesizes the peptidoglycans needed to build the bacterial cell wall [46]. Hence, finding a ligand to inhibit this protein can open the way for designing new antibacterial agents. According to the results, P5 was able to interact with the binding site residues of PBP1a, including Lys669, Thr670, Thr672, and Tyr707, with a binding energy of– 7.5 kcal/mol and inhibition constant of 5.2 μM (Fig 3c, S2 Table in S1 File). On the other hand, IARS, an aminoacyl-tRNA synthetase, catalyzes the aminoacylation of tRNA by its cognate iso-leucine which is essential for protein synthesis in bacteria, archaea, and eukaryotes [30]. There-fore, finding a ligand to inhibit it can be helpful for the inhibition of pathogens. Given this, P3 was able to interact with Gly45, Thr48, Asn50, Glu94, Gly551, and Gln554 residues of IARS with a binding energy of– 8.4 kcal/mol and inhibition constant of 1.2 μM (Fig 3d, S2 Table in S1 File). These interactions were covered the binding site of IARS through Gly551 and Gln554 residues so that P3 may act as inhibitor of IARS. *In vitro* previous literature has showed that mupirocin inhibits IARS of *Staphylococcus aureus* II-type mutant and *S. aureus* with a Ki value of 2.31 and $60 \times 10^{-3}$ μM, respectively. In comparison, P3 also revealed a significant Ki value *in silico* that can be helpful to develop new antibacterial drugs. Furthermore, the inhibition of DNA synthesis is other way to suppress the growth and development of bacteria. One of the targets for this aim is DNA gyrase which introduce the negative supercoiling into bacterial DNA [47]. Our findings showed that P4 interacts with Asn54, Glu91, Lys93, Ser128, and Thr173 residues of DNA gyrase with a binding energy of– 7.0 kcal/mol and inhibition constant of 11.8 μM (Fig 3e, S2 Table S1 File). Although P4 did not interact with the residues of the DNA gyrase active site, it is hypothesized that it may inhibit the role of DNA gyrase by disrupt-ing its function. In comparison, an *in vitro* previous work showed that TTGGQYAVAVARVVGD and LMMWVDSKNTVPKEWV peptide sequences were able to inhibit the *E. coli* DNA gyrase with a Ki value of 2.2 and 26.0 μM, respectively [48]. Also, DHFR, an metabolic enzyme for catalyzing dihydrofolic acid to tetrahydrofolic acid, has received a great deal of attention as a molecular target for bacterial resistance for decades [49]. Accordingly, Lee and coworkers showed that Trimethoprim, a selective inhibitor for bacterial DHFR, was able to bind to *Streptococcus pneumoniae* DHFR with an *in vitro* Ki value of 147 nM [50], whereas we showed that P5 can bind to *S. aureus* DHFR with an *in silico* Ki value of 0.74 μM and a binding energy of– 8.7 kcal/mol (Fig 3f, S2 Table in S1 File). Similar to DNA gyrase, P5 also interacted with residues outside DHFR active site, including Asn19, Thr47, Thr97, and Asp121, which may inhibit its function by disrupting its activity. However, our findings indi-cate that desired peptides are excellent candidates for developing new natural food preservative or drugs in the treatment of bacterial or cancerous diseases and will open new avenues for designing more selective and effective drugs. Furthermore, these peptides could recombinantly be produced in the plant expression systems to use as a fresh functional food [51, 52].

## Conclusion

The introduction of one or more clinically active candidate molecules against a disease-related target is an important goal of drug discovery. In addition, these molecules have to provide ade-quate safety and drug-like properties. Therefore, peptides derived from dietary proteins such as milk proteins, in addition to being naturally occurring, have no side effects. Hence, our

results revealed the discovery of peptides that have anti-breast cancer and antibacterial properties. Accordingly, peptides derived from whey protein with the use of both the pepsin and trypsin enzymes showed very good activity against MCF-7 breast cancer with cell viability of 7.13%. This suggests that the use of both enzymes is capable of producing peptides that exhibit high anticancer activity, whereas this treatment on casein protein produced peptides that had fewer anticancer effects than either treatment, which was used separately. In contrast, each enzyme treatment separately on whey protein showed the highest antibacterial activity against *S. aureus* and *E. coli* compared to the others. Followed by this, in the bioinformatics analysis section, we identified two highly efficient peptides that bind to specific targets in breast cancer and bacteria, one derived from casein protein (P3) and the other from whey protein (P5). This suggests that casein-derived peptides may not have been able to provide promising results *in vitro* due to their concentration distribution in the hydrolyzed mixture in compared to whey-derived peptides, although casein-derived peptides in turn have presented favorable anticancer and antibacterial activity. Therefore, our results introduce two very desirable natural food preservative or preclinical drug candidates that can be delivered to clinical and animal trials.

## Supporting information

**S1 File.**
(DOCX)

## Acknowledgments

We would like to thank Prof. Ali Niazi and Dr. Mohammad Sadegh Taghizadeh for technical supporting at Institute of Biotechnology, Shiraz University, Shiraz, Iran.

## Author Contributions

**Conceptualization:** Hamid Ezzatpanah.

**Data curation:** Mohammad Javad Taghipour.

**Formal analysis:** Mohammad Javad Taghipour, Hamid Ezzatpanah.

**Investigation:** Mohammad Javad Taghipour.

**Methodology:** Mohammad Javad Taghipour, Hamid Ezzatpanah, Mohammad Ghahderijani.

**Project administration:** Hamid Ezzatpanah, Mohammad Ghahderijani.

**Resources:** Mohammad Javad Taghipour, Hamid Ezzatpanah, Mohammad Ghahderijani.

**Software:** Mohammad Javad Taghipour, Mohammad Ghahderijani.

**Supervision:** Hamid Ezzatpanah.

**Validation:** Mohammad Javad Taghipour, Hamid Ezzatpanah.

**Visualization:** Mohammad Javad Taghipour.

**Writing – original draft:** Mohammad Javad Taghipour.

**Writing – review & editing:** Hamid Ezzatpanah.

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
