## [Decision Letter · Decision Letter 0]

19 Apr 2023

PONE-D-23-05337In vitro and in silico studies for identification bioactive peptides from camel milk protein hydrolysatesPLOS ONE

Dear Dr. Ezzatpanah,

Thank you for submitting your manuscript to PLOS ONE. After careful consideration, we feel that it has merit but does not fully meet PLOS ONE’s publication criteria as it currently stands. Therefore, we invite you to submit a revised version of the manuscript that addresses the points raised during the review process.

We look forward to receiving your revised manuscript.

Kind regards,

Ghulam Mustafa, PhD

Academic Editor

PLOS ONE

Journal Requirements:

Reviewers' comments:

Reviewer's Responses to Questions

**Comments to the Author**

1. Is the manuscript technically sound, and do the data support the conclusions?

Reviewer #1: Yes

Reviewer #2: Yes

2. Has the statistical analysis been performed appropriately and rigorously? 

Reviewer #1: Yes

Reviewer #2: Yes

3. Have the authors made all data underlying the findings in their manuscript fully available?

Reviewer #1: Yes

Reviewer #2: Yes

4. Is the manuscript presented in an intelligible fashion and written in standard English?

Reviewer #1: Yes

Reviewer #2: No

5. Review Comments to the Author

Reviewer #1: Minor modifications required:

1. Indicate the bioactivities that are being investigated in the study in your title - i.e. you can add anti-cancer and anti-bacterial bioactivities in the title.

2. Line 43: Remove the link and reference it according to the guidelines.

3. Line 80: State the storage conditions between the period of purchase and analysis.

4. Line 90: Avoid using ddH2O, instead use distilled water/double distilled water accordingly.

5. Table 4 in the reference "" ext-link-type="uri" xlink:type="simple">https://doi.org/10.1016/j.foodchem.2021.131444" has a list of camel milk-derived bioactive peptides with potential antibacterial properties. Check it out and use it in your discussion to see if there are any similar trends.

Reviewer #2: The manuscript “In vitro and in silico studies for identification bioactive peptides from camel milk protein hydrolysates” is a manuscript that investigates isolating the protein from camel milk and hydrolyzing it using two different enzymes to achieve the hydrolysate fractions with biological activities. In my opinion, the study has been well-designed, and the manuscript covers the objectives. However, some points should be carefully addressed by the authors. I believe this manuscript needs MAJOR Modification to be suitable for publishing in the journal.

Comments:

1. Title: Add “the” and “of” to the title. In vitro and in silico studies for the identification of bioactive peptides from camel milk protein hydrolysates

2. Introduction- Page 9, Line 43: use the references according to the journal requirement for web pages.

3. Materials and Methods- Page 11, Line 93: Why did the authors select 12 h for hydrolysis of the proteins?

4. Did the authors measure the DH of the parent protein as a control? If not, how do they justify that the DH of protein hydrolysates is higher or lower than the parent protein?

5. The above question is important about the bioactivities of the protein hydrolysates. How do they claim that the anticancer activity of protein hydrolysates is higher than the parent proteins? Please justify.

6. Why do authors use only 4 bacteria for the antibacterial activities? For antibacterial studies, usually, a combination of gram-negative and gram-positive bacteria should be used. Justify please.

6. PLOS authors have the option to publish the peer review history of their article (what does this mean?). If published, this will include your full peer review and any attached files.

Reviewer #1: No

Reviewer #2: No

---

## [Author Response · Author response to Decision Letter 0]

27 May 2023

Reviewer #1: Minor modifications required:

1. Indicate the bioactivities that are being investigated in the study in your title - i.e. you can add anti-cancer and anti-bacterial bioactivities in the title.

Response: Dear Reviewer, we would like to thank you for your time and for helping improve our manuscript with your valuable comments.

Title was revised as you wished.

2. Line 43: Remove the link and reference it according to the guidelines.

Response: we revised it according to the journal guideline.

3. Line 80: State the storage conditions between the period of purchase and analysis.

Response: we added it in the main text (line 81).

4. Line 90: Avoid using ddH2O, instead use distilled water/double distilled water accordingly.

Response: we revised it as you wished.

5. Table 4 in the reference "https://doi.org/10.1016/j.foodchem.2021.131444" has a list of camel milk-derived bioactive peptides with potential antibacterial properties. Check it out and use it in your discussion to see if there are any similar trends.

Response: Thank you for your good suggestion. We included it in the discussion (lines 221-228).

Reviewer #2: The manuscript "In vitro and in silico studies for identification bioactive peptides from camel milk protein hydrolysates" is a manuscript that investigates isolating the protein from camel milk and hydrolyzing it using two different enzymes to achieve the hydrolysate fractions with biological activities. In my opinion, the study has been well-designed, and the manuscript covers the objectives. However, some points should be carefully addressed by the authors. I believe this manuscript needs MAJOR Modification to be suitable for publishing in the journal.

Comments:

1. Title: Add "the" and "of" to the title. In vitro and in silico studies for the identification of bioactive peptides from camel milk protein hydrolysates

Response: Dear Reviewer, we would like to thank you for your time and for helping improve our manuscript with your valuable comments.

Title was edited as you wished. 

2. Introduction- Page 9, Line 43: use the references according to the journal requirement for web pages.

Response: we revised it according to the journal guideline.

3. Materials and Methods- Page 11, Line 93: Why did the authors select 12 h for hydrolysis of the proteins?

Response: This time was chosen due to (i) Ensure the complete hydrolysis of the protein by the enzyme, that is, the enzyme has the opportunity to recognize and cut all its recognition sites (ii) Since our aim was to evaluate the antibacterial and anticancer activities, therefore we wanted to ensure the production of 1-7 kD peptides because the peptides with these sizes usually exhibit these activities (https://doi.org/10.1098/rsob.200004; https://doi.org/10.1128/AAC.02369-13). Also, in a study by Wang et al. (https://doi.org/10.3390/ani10020337), they stated that hydrolysis of camel milk proteins for 12h could produce peptides for killing their desired bacteria (lines 228-232).

4. Did the authors measure the DH of the parent protein as a control? If not, how do they justify that the DH of protein hydrolysates is higher or lower than the parent protein?

Response: Thank you for your precious question. At time 0, no hydrolysis has taken place, so this in turn represents the degree of hydrolysis of the original proteins initially. However, we added the following sentence: “At time 0, no hydrolysis has taken place, so that the degree of hydrolysis of the parent proteins varied from 1.9 to 2.8% and increased over time with the addition of enzymes (Fig. 1)" in lines 195-197 to avoid any misunderstanding.

5. The above question is important about the bioactivities of the protein hydrolysates. How do they claim that the anticancer activity of protein hydrolysates is higher than the parent proteins? Please justify.

Response: According to the Fig. 2, we measured anticancer activity of parent proteins and hydrolysates, therefore whey and casein proteins (parent proteins that used as control) respectively killed approximately 30% and 10% of total cancerous cells, which is lower than anticancer activity of hydrolysates (Fig. 2). Also, it is better to say that since we boiled the mixture after hydrolysis to inactivate the enzyme, all of proteins, including enzyme and parent proteins were removed using centrifugation. Therefore, there are only peptides in the mixture. But if there are parent proteins in the mixture, they are inactive. However, we clarified it by adding a sentence in lines 250-252.

6. Why do authors use only 4 bacteria for the antibacterial activities? For antibacterial studies, usually, a combination of gram-negative and gram-positive bacteria should be used. Justify please.

Response: Thank you so much for your good question. As you said we also used a combination of gram-negative and gram-positive bacteria (3 gram-negatives bacteria, including Klebsiella pneumoniae (PTCC 1053), Pseudomonas aeruginosa (PTCC 1074), and Escherichia coli (PTCC 1330); and 1 gram-positive bacteria, including Staphylococcus aureus (PTCC 1112)). They are from different families and commonly use in antibacterial studies, because they cause deadly diseases in humans. Therefore, we decided to choose them for evaluating antibacterial tests.

---

## [Decision Letter · Decision Letter 1]

22 Jun 2023

In vitro and in silico studies for the identification of anti-cancer and antibacterial peptides from camel milk protein hydrolysates

PONE-D-23-05337R1

Dear Dr. Ezzatpanah,

We’re pleased to inform you that your manuscript has been judged scientifically suitable for publication and will be formally accepted for publication once it meets all outstanding technical requirements.

Kind regards,

Ghulam Mustafa, PhD

Academic Editor

PLOS ONE

Additional Editor Comments (optional):

Reviewers' comments:

Reviewer's Responses to Questions

**Comments to the Author**

1. If the authors have adequately addressed your comments raised in a previous round of review and you feel that this manuscript is now acceptable for publication, you may indicate that here to bypass the “Comments to the Author” section, enter your conflict of interest statement in the “Confidential to Editor” section, and submit your "Accept" recommendation.

Reviewer #1: All comments have been addressed

Reviewer #2: All comments have been addressed

2. Is the manuscript technically sound, and do the data support the conclusions?

Reviewer #1: Yes

Reviewer #2: Yes

3. Has the statistical analysis been performed appropriately and rigorously? 

Reviewer #1: Yes

Reviewer #2: Yes

4. Have the authors made all data underlying the findings in their manuscript fully available?

Reviewer #1: Yes

Reviewer #2: Yes

5. Is the manuscript presented in an intelligible fashion and written in standard English?

Reviewer #1: Yes

Reviewer #2: Yes

6. Review Comments to the Author

Reviewer #1: All comments have been addressed. The manuscript can be accepted in its current form. Thanks for the revision.

Reviewer #2: (No Response)

7. PLOS authors have the option to publish the peer review history of their article (what does this mean?). If published, this will include your full peer review and any attached files.

Reviewer #1: No

Reviewer #2: No

---

## [Editor Report · Acceptance letter]

3 Jul 2023

PONE-D-23-05337R1 

*In vitro* and *in silico* studies for the identification of anti-cancer and antibacterial peptides from camel milk protein hydrolysates 

Dear Dr. Ezzatpanah:

I'm pleased to inform you that your manuscript has been deemed suitable for publication in PLOS ONE. Congratulations! Your manuscript is now with our production department. 

Kind regards, 

on behalf of

Dr. Ghulam Mustafa 

Academic Editor

PLOS ONE